# Finite and Corruption-Robust Regret Bounds in Online Inverse Linear Optimization under M-Convex Action Sets

**Taihei Oki** [* 1 2 3]    **Shinsaku Sakaue** [* 4 5 3]

## Abstract

We study online inverse linear optimization, also known as contextual recommendation, where a *learner* sequentially infers an *agent*'s hidden objective vector from observed optimal actions over feasible sets that change over time. The learner aims to recommend actions that perform well under the agent's true objective, and the performance is measured by the *regret*, defined as the cumulative gap between the agent's optimal values and those achieved by the learner's recommended actions. Prior work has established a regret bound of $O(d \log T)$, as well as a finite but exponentially large bound of $\exp(O(d \log d))$, where $d$ is the dimension of the optimization problem and $T$ is the time horizon, while a regret lower bound of $\Omega(d)$ is known (Gollapudi et al. 2021; Sakaue et al. 2025). Whether a finite regret bound polynomial in $d$ is achievable or not has remained an open question. We partially resolve this by showing that when the feasible sets are *M-convex*—a broad class that includes matroids—a finite regret bound of $O(d \log d)$ is possible. We achieve this by combining a structural characterization of optimal solutions on M-convex sets with a geometric volume argument. Moreover, we extend our approach to adversarially corrupted feedback in up to $C$ rounds. We obtain a regret bound of $O((C + 1)d \log d)$ without prior knowledge of $C$, by monitoring directed graphs induced by the observed feedback to detect corruptions adaptively.

---

[*]Equal contribution  [1]Institute for Chemical Reaction Design and Discovery (ICReDD), Hokkaido University, Sapporo, Hokkaido, Japan [2]D3 Center, The University of Osaka, Osaka, Japan [3]Center for Advanced Intelligence Project, RIKEN, Tokyo, Japan [4]CyberAgent, Tokyo, Japan [5]National Institute of Informatics, Tokyo, Japan. Correspondence to: Taihei Oki <oki@icredd.hokudai.ac.jp>, Shinsaku Sakaue <shinsaku.sakaue@gmail.com>.

*Proceedings of the 43rd International Conference on Machine Learning*, Seoul, South Korea. PMLR 306, 2026. Copyright 2026 by the author(s).

## 1. Introduction

Inverse optimization (Ahuja and Orlin, 2001; Chan et al., 2025) is the problem of estimating parameters of optimization problems from observed optimal solutions. Since its early development in seismic tomography (Tarantola, 1988; Burton and Toint, 1992), inverse optimization has been applied to a wide range of tasks, such as inferring customer preferences from purchase behavior (Bärmann et al., 2017, 2020) and identifying market structures in the electricity market (Birge et al., 2017). In this paper, we focus on inverse optimization with linear objectives, a fundamental setting that is both theoretically rich and practically relevant.

In many realistic scenarios, optimal solutions are observed sequentially, which naturally leads to the study of online inverse linear optimization. Bärmann et al. (2017, 2020) initiated the study of such an online framework, where an *agent* sequentially takes actions that are optimal solutions to linear optimization problems of the form

$$\text{maximize } \langle w^*, x \rangle \qquad \text{subject to } x \in X_t \qquad (1)$$

for $t = 1, \ldots, T$. Here, $T$ is the time horizon, $d$ is the dimension of the problem, $w^* \in \mathbb{R}^d$ is the agent's hidden objective vector, and $X_t \subseteq \mathbb{R}^d$ is a feasible set at round $t$. A *learner* makes an estimate $\hat{w}_t \in \mathbb{R}^d$ of the agent's objective vector at each round $t$ based on observations of the agent's past actions and the feasible sets. A natural performance metric is the *regret*, which measures the cumulative gap between the optimal objective values and those achieved by the actions chosen based on the learner's estimates.

Bärmann et al. (2017, 2020) obtained a regret upper bound of $O(\sqrt{T})$. Subsequently, Besbes et al. (2021, 2025), Gollapudi et al. (2021), Sakaue et al. (2025b), and Sakaue (2026) obtained regret upper bounds that depend only logarithmically on $T$. Sakaue et al. (2025b) also give a lower bound of $\Omega(d)$, which is intuitive since the learner must estimate $d$ parameters of the agent's objective vector. Table 1 summarizes these results.

However, those regret upper bounds still depend at least logarithmically on $T$, which can be non-negligible in applications where observations accumulate rapidly. Gollapudi et al. (2021) also obtained a regret bound of $\exp(O(d \log d))$.

*Table 1.* Regret bounds for online inverse linear optimization (contextual recommendation). Here, we treat the sizes of the agent's feasible sets and the learner's decision set as constants; see Sakaue et al. (2025b, Appendix A) for a discussion on how these sizes affect the regret bounds. The column "Corruption" indicates whether corrupted feedback is considered, while the way of handling corruption differs among the works; see Section 2.2 for a discussion.

| | Reference | Regret Bound | Corruption | Note |
|---|---|---|---|---|
| Upper | Bärmann et al. (2017, 2020) | $O(\sqrt{T})$ | ✓ | |
| | Besbes et al. (2021, 2025) | $O(d^4 \log T)$ | | |
| | Gollapudi et al. (2021) | $O(d \log T), \exp(O(d \log d))$ | | |
| | Sakaue et al. (2025b), Sakaue (2026) | $O(d \log T)$ | ✓ | |
| | Sakaue et al. (2025a) | $O(1/\Delta^2)$ | | Under the $\Delta$-gap condition |
| | This work | $O(d \log d)$ | ✓ | Under M-convex action sets |
| Lower | Sakaue et al. (2025b) | $\Omega(d)$ | | Thm. 6.1 for the M-convex case |

While this bound is finite (independent of $T$), its exponential dependence on the dimension $d$ severely limits its practical usefulness. Sakaue et al. (2025b) showed that if the true objective vector is distant from the decision boundary by $\Delta$ over all rounds, a regret upper bound of $O(1/\Delta^2)$ can be achieved; however, in general, decision boundaries at some rounds may become very close to the true objective vector, particularly when $T$ is huge. Therefore, achieving a finite and practically meaningful regret bound—especially one that scales polynomially in $d$—remains an important open problem.

## 1.1. Our Contribution

We show that when the feasible sets are *M-convex*, a finite regret bound of $O(d \log d)$ is achievable. This result partially resolves the question left open by Besbes et al. (2021, 2025), Gollapudi et al. (2021), Sakaue et al. (2025b), and Sakaue (2026) regarding the existence of a finite regret upper bound polynomial in $d$, under M-convex action sets. M-convex sets subsume *matroid* action sets, a broad class of combinatorial structures that arise in many settings and are important both theoretically and practically; see also Section 2.3 for details. At a technical level, our algorithm combines a characterization of optimal solutions on M-convex sets (Proposition 2.4) with a geometric volume argument (Lemma 4.1). This strategy for deriving the finite regret bound highlights the usefulness of M-convexity in online inverse linear optimization.

Furthermore, we address scenarios where the agent's actions may be adversarially corrupted in up to $C$ rounds, thereby relaxing the restrictive assumption that the agent's action is always optimal. For this setting, we extend our algorithm by combining it with a restart mechanism and obtain a regret upper bound of $O((C + 1)d \log d)$,[1] which recovers the original bound when there is no corruption and offers a robust guarantee in the presence of corruptions. Importantly,

[1]We write $O((C+1)d \log d)$ instead of $O(Cd \log d)$ to clarify that the bound holds even when there is no corruption ($C = 0$).

our algorithm does not require prior knowledge of $C$. We achieve this by monitoring the acyclicity of directed graphs constructed from the agent's feedback (Lemma 5.1).

Finally, we show that the proof of Sakaue et al. (2025b) for the regret lower bound of $\Omega(d)$ can be adapted to our M-convex case, establishing the same lower bound even when the agent's feasible sets are M-convex. Therefore, our regret upper bound of $O(d \log d)$ is tight up to a $\log d$ factor.

Below we summarize our contributions for online inverse linear optimization under M-convex action sets.

- We present an algorithm with a regret upper bound of $O(d \log d)$, partially resolving the open question regarding the existence of a finite regret bound polynomial in $d$.

- We extend the algorithm to handle adversarial corruptions in up to $C$ rounds, achieving a regret upper bound of $O((C + 1)d \log d)$ without prior knowledge of $C$.

- We give a regret lower bound of $\Omega(d)$ for the M-convex case, showing that our $O(d \log d)$ upper bound is tight up to a $\log d$ factor.

**Motivation for M-Convex Action Sets.** Before moving to the main part, we discuss why M-convex action sets are of interest. When the feasible set is curved, such as an ellipsoid, the inverse problem becomes trivial because observing a single optimal solution already pins down the direction of the objective vector—the normal cone collapses to a single ray. The genuinely challenging case is when the action sets are non-curved, such as polyhedral sets, which typically arise in combinatorial optimization. M-convex sets form a broad subclass of such action sets with useful combinatorial structure; see Section 2.3 for examples. Focusing on M-convex sets therefore lets us cover a wide range of practically relevant problems while retaining enough structure to establish improved regret bounds.

**Organization.** This paper is organized as follows. The rest of this section discusses related work. Section 2 presents

the problem setting and preliminaries. As a warm-up, Section 3 gives a basic $O(d^2)$-regret algorithm, and Section 4 refines it to achieve the $O(d \log d)$ regret bound. Section 5 extends the algorithm to handle corruptions. Section 6 gives the $\Omega(d)$ regret lower bound. Section 7 concludes the paper.

### 1.2. Related Work

As noted in Section 1, inverse optimization has long been used to infer hidden parameters from observed optimal solutions. Recent work has been particularly active in data-driven offline settings that estimate parameters from multiple observed solutions (Bertsimas et al., 2015; Aswani et al., 2018; Mohajerin Esfahani et al., 2018). For the online setting, Bärmann et al. (2017) initiated a line of work on sequentially learning from optimal actions. This line has led to regret bounds summarized in Table 1; see also Sakaue et al. (2025b) for an overview.

Another related but distinct line of research is contextual search/pricing (Liu et al., 2021; Paes Leme and Schneider, 2022), where the learner aims to infer $\langle w^*, x_t \rangle$ from context vectors $x_t$ under limited feedback. Gollapudi et al. (2021) leverage techniques from this literature for contextual recommendation, which is equivalent to online inverse linear optimization up to terminology. Our analysis also borrows techniques from this literature, particularly the volume argument in Section 4, while our novelty lies in synthesizing this argument with the M-convex structure to achieve the finite regret bound of $O(d \log d)$. Corruption-robust methods have also attracted attention in this field (Krishnamurthy et al. 2021; Paes Leme et al. 2022, 2025; Gupta et al. 2025). In particular, Gupta et al. (2025) have proposed, for contextual pricing, a framework that runs $C+1$ parallel copies of an uncorrupted algorithm to obtain a regret bound of $O(C)$ times the uncorrupted regret. Our method yields a similar multiplicative $O(C)$ degradation but follows a different route: it leverages the M-convex structure to detect corruptions and uses restarts instead of maintaining multiple copies.

M-convexity is a fundamental concept in *discrete convex analysis* (Murota, 2003), which arises in various fields, such as resource allocation (Katoh et al., 2013) and economics (Murota, 2022). Recently, M-convex function minimization has gained attention in the area of algorithms with predictions (Oki and Sakaue, 2023) and online learning (Oki and Sakaue, 2024), but the role of M-convexity in online inverse linear optimization has been unexplored.

## 2. Preliminaries

**Notation.** Let $[d] = \{1, \ldots, d\}$ for any positive integer $d$. For a vector $x \in \mathbb{R}^d$ and $i \in [d]$, let $x(i)$ denote the $i$th component. For $i \in [d]$, we use $\mathbf{e}_i \in \mathbb{R}^d$ to denote the $i$th standard basis vector, i.e., $\mathbf{e}_i(j) = 1$ if $i = j$ and $0$ otherwise.

### 2.1. Problem Setting

For $t = 1, \ldots, T$, the agent, who has a hidden objective vector $w^* \in \mathbb{R}^d$, solves (1) and takes the obtained action $x_t \in X_t$. In each round $t$, the learner computes an estimate $\hat{w}_t \in \mathbb{R}^d$ based on the information $\{(X_s, x_s)\}_{s=1}^{t-1}$ observed up to round $t-1$. Then, given the feasible set $X_t$, we define the action suggested by the learner's estimate $\hat{w}_t$ as

$$\hat{x}_t \in \arg\max\{\, \langle \hat{w}_t, x \rangle \,:\, x \in X_t \,\}.$$

Note that, at this point, $X_t$ is used solely to define $\hat{x}_t$; the learner's estimate $\hat{w}_t$ is computed from information up to round $t-1$. Then, the learner observes the agent's actual action $x_t$ and the feasible set $X_t$, which will be used to compute the next estimate $\hat{w}_{t+1}$. In what follows, we suppose that the learner knows $X_t$ at the beginning of round $t$ for simplicity, although our analysis applies to the case where $X_t$ is revealed only after the learner chooses $\hat{w}_t$.

The learner's regret is defined as

$$R_T = \sum_{t=1}^{T} \langle w^*, x_t - \hat{x}_t \rangle, \tag{2}$$

which is the cumulative gap between the optimal values and objective values achieved by following the learner's choices. In this study, we focus on the case where the agent's feasible sets $X_t \subseteq \mathbb{Z}^d$ (embedded in $\mathbb{R}^d$) are M-convex; see Section 2.3 for details.

**Remark 2.1** (On the regret metric). Although our regret notion differs from the standard inverse-optimization goal of fully estimating $w^*$, it is the natural online analogue of the *estimation loss* widely used in inverse optimization (Chen and Kılınç-Karzan, 2020; Sun et al., 2023; Sakaue et al., 2025a). Moreover, because we only observe optimal actions (and feasible sets), components of $w^*$ that do not affect the choice of optimal actions cannot be inferred accurately. In this light, the regret based on the gap between observed and predicted actions is an appropriate performance measure.

For simplicity, we make the following assumption.

**Assumption 2.2.** *We assume the following conditions.*

1. $\max\{\, \langle w^*, x - x' \rangle \,:\, x, x' \in X_t \,\} = O(1)$ *for all $t$.*

2. *The components of $w^*$ are distinct.*

The first condition means that the per-round regret is $O(1)$, which is a standard simplification. Note that as the agent's objective function is linear, we are only interested in estimating the direction of $w^*$, not in its magnitude; therefore, we can assume any bounds on the magnitude of $w^*$ for convenience. The second condition is mainly to avoid complications arising from ties among components of $w^*$. As detailed in Appendix D, this condition can indeed be removed in the uncorrupted case (Sections 3 and 4), while it is needed in the corrupted case (Section 5).

## 2.2. Corrupted Feedback

Our corruption model follows the standard one in the literature on contextual search/pricing (Krishnamurthy et al. 2021; Paes Leme et al. 2022, 2025; Gupta et al. 2025). Specifically, we define the corruption level $C$ as the number of rounds $t$ in which the agent's action $x_t \in X_t$ is not an optimal solution to (1), i.e.,

$$C = \left| \left\{ t \in [T] : x_t \notin \arg\max_{x \in X_t} \langle w^*, x \rangle \right\} \right|.$$

We do not assume that the learner knows $C$ in advance.

The regret in this setting is still defined by (2), using the actual actions $x_t$ taken by the agent. One might consider defining the regret using (unobserved) optimal actions as

$$R_T^* = \sum_{t=1}^T \langle w^*, x_t^* - \hat{x}_t \rangle \quad \text{where} \quad x_t^* \in \arg\max_{x \in X_t} \langle w^*, x \rangle.$$

Since $\langle w^*, x_t^* - x_t \rangle$ becomes positive at most $C$ times, we have $R_T^* \lesssim R_T + C$. Also, $R_T$ is $\Omega(C)$ in the worst case. Therefore, the two regret definitions admit essentially the same bounds up to constant factors.

Sakaue et al. (2025b) and Sakaue (2026) study a more flexible framework where corruption is measured by the cumulative suboptimality of the corrupted actions, but their corruption-robust regret bound still includes a $\log T$ factor as in their uncorrupted case. Extending our approach to their corruption framework while retaining a finite regret bound is an interesting direction for future work.

## 2.3. M-Convex Set

M-convex sets are a central notion in discrete convex analysis (Murota, 2003), which is characterized by an exchange property over the integer lattice as follows.

**Definition 2.3** (Murota 2003, Section 4.1)**.** A non-empty set of integer points, $X \subseteq \mathbb{Z}^d$, is *M-convex* if the following condition holds: for any $x, y \in X$ and $i \in [d]$ with $x(i) > y(i)$, there exists $j \in [d]$ with $x(j) < y(j)$ such that

$$x - \mathbf{e}_i + \mathbf{e}_j \in X \quad \text{and} \quad y + \mathbf{e}_i - \mathbf{e}_j \in X.$$

An important property of M-convex sets is the following characterization of optimal solutions to linear optimization over M-convex sets.

**Proposition 2.4** (Murota 2003, Theorem 6.26)**.** *Let $X \subseteq \mathbb{Z}^d$ be an M-convex set. For any $w \in \mathbb{R}^d$ and $x \in X$, the following two conditions are equivalent:*

*1. $x \in \arg\max\{ \langle w, x' \rangle : x' \in X \}$.*

*2. $w(i) \geq w(j)$ for all $i, j \in [d]$ with $x - \mathbf{e}_i + \mathbf{e}_j \in X$.*

That is, an action $x$ is optimal for $w$ if and only if no local exchange of one unit between any two indices $i$ and $j$ that maintains the feasibility can increase the objective value. Intuitively, this property enables us to gain more information about the objective vector $w^*$ from observed optimal solutions compared to the general case.

This useful structure indeed appears in various interesting problems. Below we present such examples.

**Example 1: Matroids.** Let $\mathcal{B} \subseteq 2^{[d]}$ be a set family over $[d]$. We say $([d], \mathcal{B})$ is a *matroid* if $\mathcal{B}$ is non-empty and satisfies the exchange property, i.e., for any $B_1, B_2 \in \mathcal{B}$ and $i \in B_1 \setminus B_2$, there exists $j \in B_2 \setminus B_1$ such that

$$(B_1 \setminus \{i\}) \cup \{j\} \in \mathcal{B} \quad \text{and} \quad (B_2 \setminus \{j\}) \cup \{i\} \in \mathcal{B}.$$

We call each element of $\mathcal{B}$ a *basis*. By writing this in terms of characteristic vectors, i.e., identifying each basis $B \in \mathcal{B}$ with its characteristic vector $\chi_B \in \{0,1\}^d$ defined as $\chi_B(i) = 1$ if $i \in B$ and 0 otherwise, the exchange property of matroids coincides with that of M-convex sets on $\{0,1\}^d$. Therefore, matroids form a subclass of M-convex sets. This observation allows us to handle various situations where problem (1) is combinatorial optimization over matroids, such as maximization over $m$-sets (subsets of size $m \leq d$) and spanning trees in graphs, which correspond to uniform and graphic matroids, respectively.

**Example 2: Extensions to integer lattice.** Let $D \in \mathbb{Z}_{>0}$ and $m \in [dD]$. Consider a situation where the agent is given $d$ kinds of items, each of which can be chosen up to $D$ units, and the total number of chosen items must be $m$.[2] Then, the agent's feasible set is given as

$$X = \left\{ x \in \{0, 1, \ldots, D\}^d : \sum_{i=1}^d x(i) = m \right\}.$$

This can be viewed as an extension of $m$-sets to the integer lattice, and satisfies the definition of M-convex sets. Note that naively considering $D$ copies of each item increases the dimension to $dD$ and worsens the regret bounds.[3] Similarly, we can naturally extend other matroids to M-convex sets; for example, we can extend the partition (or laminar) matroid by allowing multiple copies of each element.

---

[2] We can focus on the equality constraint when components of $w^*$ are non-negative without loss of generality; otherwise, the feasible set defined by the inequality constraint is $M^\natural$-*convex* in $\mathbb{Z}^d$, which is indeed an M-convex set in $\mathbb{Z}^{d+1}$ (Murota, 2003, Section 4.3) and hence can be handled by our framework.

[3] Strictly speaking, defining $X$ on the domain $\{0, 1, \ldots, D\}^d$ leads to a regret bound scaled up by a factor of $D$ compared to the alternative one derived by defining $X$ on $\{0, 1\}^{dD}$. Nonetheless, regret bounds that are superlinear in the dimension become larger when the dimension increases to $dD$.

---

**Algorithm 1** Algorithm for uncorrupted feedback

---

1: Let $A_1 = \emptyset$.
2: **for** $t = 1, 2, \ldots, T$ **:**
3:      Take $\hat{w}_t$ with distinct components such that
         $\hat{w}_t(i) > \hat{w}_t(j)$    for all $(i, j) \in A_t$.
4:      Let $\hat{x}_t \in \arg\max\{ \langle \hat{w}_t, x \rangle : x \in X_t \}$.
5:      Observe the agent's action $x_t$.
6:      $A_{t+1} \leftarrow A_t \cup \{ (i, j) : i \neq j, x_t - \mathbf{e}_i + \mathbf{e}_j \in X_t \}$.

---

## 3. Warm-up: $O(d^2)$-Regret Algorithm

We begin by presenting a simple algorithm that achieves a regret upper bound of $O(d^2)$ based on Proposition 2.4. This and the next sections assume that the agent's actions are uncorrupted, i.e., $x_t \in \arg\max_{x \in X_t} \langle w^*, x \rangle$ holds for all $t$.

Algorithm 1 shows the details. The set $A_t$ maintains pairs of indices $(i, j)$ such that $w^*(i) > w^*(j)$ must hold for past observations $x_1, \ldots, x_{t-1}$ to be optimal for $w^*$; note that $A_t \subseteq A_{t+1}$ always holds. Furthermore, the directed graph defined by $([d], A_t)$ is always acyclic. Indeed, if it contained a directed cycle $i_1 \to i_2 \to \cdots \to i_k \to i_1$, then, we would have $w^*(i_1) > w^*(i_2) > \cdots > w^*(i_k) > w^*(i_1)$, as the components of $w^*$ are distinct, which is a contradiction. Therefore, we can always take $\hat{w}_t$ that satisfies the condition in Step 3, i.e., $\hat{w}_t(i) > \hat{w}_t(j)$ for all $(i, j) \in A_t$, by performing a topological sort of this acyclic graph. The notion of the directed graph $([d], A_t)$ and its acyclicity property will also play a key role in Section 5 for handling corrupted feedback.

The regret of Algorithm 1 is bounded as follows.

**Theorem 3.1.** *Algorithm 1 achieves*

$$R_T = O(d^2).$$

*Proof.* We prove the claim by showing that $x_t \neq \hat{x}_t$ can happen $O(d^2)$ times. In round $t$, if $A_{t+1} = A_t$, we have

$$\{ (i, j) \in [d] \times [d] : i \neq j, x_t - \mathbf{e}_i + \mathbf{e}_j \in X_t \} \subseteq A_t.$$

By the choice of $\hat{w}_t$ in Step 3, this means

$$\hat{w}_t(i) > \hat{w}_t(j) \quad \forall (i, j) \text{ s.t. } i \neq j, x_t - \mathbf{e}_i + \mathbf{e}_j \in X_t.$$

In addition, $\hat{w}_t(i) = \hat{w}_t(j)$ obviously holds if $i = j$. Therefore, Proposition 2.4 ensures

$$x_t \in \arg\max\{ \langle \hat{w}_t, x \rangle : x \in X_t \}.$$

Since components of $\hat{w}_t$ are distinct, the maximizer over $X_t$ is unique, i.e., $\arg\max\{ \langle \hat{w}_t, x \rangle : x \in X_t \} = \{\hat{x}_t\}$, which can be confirmed as follows: if, for contradiction, there are distinct maximizers $\hat{x}, \hat{y} \in \arg\max\{ \langle \hat{w}_t, x \rangle : x \in X_t \}$, the definition of M-convex sets (Definition 2.3) implies that there exist $i, j \in [d]$ such that

$$\hat{x} - \mathbf{e}_i + \mathbf{e}_j \in X_t \quad \text{and} \quad \hat{y} + \mathbf{e}_i - \mathbf{e}_j \in X_t,$$

one of which attains a larger objective value than $\hat{x}$ and $\hat{y}$ as $\hat{w}_t(i) \neq \hat{w}_t(j)$, contradicting the optimality of $\hat{x}$ and $\hat{y}$.

Consequently, whenever $A_{t+1} = A_t$, we have $x_t = \hat{x}_t$ and thus $\langle w^*, x_t - \hat{x}_t \rangle = 0$. From $A_t \subseteq A_{t+1}$ and $|A_{T+1}| \leq \binom{d}{2}$, $A_{t+1} \neq A_t$ can happen $O(d^2)$ times. Therefore, we have $\sum_t \langle w^*, x_t - \hat{x}_t \rangle = O(d^2)$, completing the proof. □

The core idea is: we incur a non-zero regret only when we mispredict the order of some index pair $(i, j)$ that matters for selecting an optimal action from $X_t$; once that happens, we can learn the correct order of $(i, j)$ and never mispredict it again. Therefore, the total number of such mistakes is at most the number of pairs of indices, which is $\binom{d}{2} = O(d^2)$. This argument is justified by the M-convexity of the action sets through Proposition 2.4.

## 4. $O(d \log d)$-Regret Algorithm

We have observed that a simple algorithm (Algorithm 1) can achieve a regret upper bound of $O(d^2)$ by maintaining pairwise orderings of indices. This section refines the algorithm to achieve a regret upper bound of $O(d \log d)$. The room for improvement lies in the arbitrariness of the choice of $\hat{w}_t$. Here, we take $\hat{w}_t$ as the center of gravity of

$$P_t = \{ w \in [0, 1]^d : w(i) \geq w(j) \text{ for all } (i, j) \in A_t \},$$

where tied components, if any, are broken by an arbitrarily small perturbation within $P_t$.[4] This choice allows us to bound the number of mistakes more tightly via a volume argument. The following lemma, derived from Grünbaum's theorem (Grünbaum, 1960), plays a key role in the analysis.

**Lemma 4.1.** *In Algorithm 1 with the above choice of $\hat{w}_t$, if $x_t \neq \hat{x}_t$, we have*

$$\mathrm{Vol}(P_{t+1}) \leq (1 - 1/e)\mathrm{Vol}(P_t),$$

*where $\mathrm{Vol}(\cdot)$ is the volume defined by the Lebesgue measure.*

*Proof.* We first show that there exists $(i, j) \in A_{t+1}$ such that $\hat{w}_t(i) < \hat{w}_t(j)$; otherwise, $\hat{w}_t(i) \geq \hat{w}_t(j)$ holds for all $(i, j) \in A_{t+1} \supseteq \{ (i, j) : i \neq j, x_t - \mathbf{e}_i + \mathbf{e}_j \in X_t \}$, and thus Proposition 2.4 implies $x_t \in \arg\max_{x \in X_t} \langle \hat{w}_t, x \rangle = \{\hat{x}_t\}$,[5] contradicting $x_t \neq \hat{x}_t$. Additionally, such $(i, j)$ must not belong to $A_t$ since we have $\hat{w}_t(i) < \hat{w}_t(j)$.

Let $(i, j) \in A_{t+1} \setminus A_t$ be such a pair. Define the halfspace

$$H = \{ w \in \mathbb{R}^d : w(i) \geq w(j) \}.$$

---

[4] A sufficiently small perturbation does not affect the constant-factor volume shrinkage in the subsequent analysis; see approximate Grünbaum's theorem (Bertsimas and Vempala, 2004).

[5] The uniqueness of the maximizer follows from the distinctness of components of $\hat{w}_t$, as argued in the proof of Theorem 3.1.

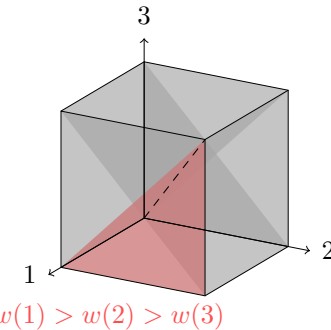

3

1

2

$w(1) > w(2) > w(3)$

*Figure 1.* The decomposition of $[0,1]^3$ into six order simplices. The order simplex for $w(1) > w(2) > w(3)$ is highlighted in red.

By the choice of $(i,j)$, we have $\hat{w}_t \in P_t \setminus H$. By Grünbaum's theorem (Grünbaum, 1960), when a convex body is cut by a hyperplane, the fraction containing the center of gravity has volume at least $1/\mathrm{e}$ of the original convex body. As the center of gravity $\hat{w}_t$ (up to an arbitrarily small perturbation) lies in $P_t \setminus H$, we have $\mathrm{Vol}(P_t \setminus H) \geq \frac{1}{\mathrm{e}}\mathrm{Vol}(P_t)$.

By the definition of $P_{t+1}$, any $w \in P_{t+1}$ satisfies $w(i) \geq w(j)$ for the chosen $(i,j) \in A_{t+1} \setminus A_t$, and thus $P_{t+1}$ is contained in $P_t \cap H$. Therefore, we have

$$\mathrm{Vol}(P_t) = \mathrm{Vol}(P_t \cap H) + \mathrm{Vol}(P_t \setminus H)$$
$$\geq \mathrm{Vol}(P_{t+1}) + \frac{1}{\mathrm{e}}\mathrm{Vol}(P_t).$$

Rearranging the terms yields the claim. □

**Theorem 4.2.** *Algorithm 1 with the choice of $\hat{w}_t$ as the center of gravity of $P_t$ (with tie-breaking) achieves*

$$R_T = O(d \log d).$$

*Proof.* By definition of $P_t$, $\mathrm{Vol}(P_t)$ is non-increasing in $t$, and Lemma 4.1 ensures that it decreases by a factor of at least $1 - 1/\mathrm{e}$ whenever $x_t \neq \hat{x}_t$. Therefore, the number of rounds we can incur a non-zero regret is at most

$$\log_{\frac{\mathrm{e}}{\mathrm{e}-1}}\left(\frac{\mathrm{Vol}(P_1)}{\mathrm{Vol}(P_T)}\right),$$

where $P_1 = [0,1]^d$ since $A_1 = \emptyset$. Below, we upper bound $\mathrm{Vol}(P_1)/\mathrm{Vol}(P_T)$.

Note that for all $(i,j) \in A_T$ we have $w^*(i) > w^*(j)$. Since the constraints defining $P_T$ depend only on these pairwise orderings, any vector $w \in [0,1]^d$ whose components have the same order as $w^*$ satisfies $w(i) \geq w(j)$ for all $(i,j) \in A_T$, and hence belongs to $P_T$. The set of such $w$ forms an order simplex, and the unit hypercube $[0,1]^d$ can be decomposed into $d!$ such order simplices of equal volume; see Figure 1.[6]

---

[6]This decomposition is classically known as the Freudenthal triangulation of the unit cube (Freudenthal, 1942) (see, e.g., Edelsbrunner and Kerber 2012 for a modern reference). We simply refer to each simplex in this triangulation as an order simplex.

Therefore, we have $\mathrm{Vol}(P_T) \geq \frac{1}{d!}\mathrm{Vol}([0,1]^d)$, and hence the number of rounds we incur a non-zero regret is at most

$$\log_{\frac{\mathrm{e}}{\mathrm{e}-1}} d! = O(d \log d),$$

completing the proof. □

**Time complexity.** The per-round time complexity of Algorithm 1 is dominated by three parts: taking $\hat{w}_t$ in Step 3, computing $\hat{x}_t$ in Step 4, and updating $A_{t+1}$ in Step 6. The complexity of Steps 4 and 6 depends on the structure of M-convex sets $X_t$, but these can be done in polynomial time given access to a membership oracle for $X_t$. Regarding Step 3, the exact computation of the center of gravity is #P-hard, even in our case where the polytope consists of order simplices (Brightwell and Winkler, 1991). Still, we can use a randomized algorithm (Lovász and Vempala, 2006) to obtain an approximate center of gravity in polynomial time, as is commonly done in the literature; see also Feldman et al. (2015, Section 5.2.1). When it comes to the $O(d^2)$-regret algorithm in Section 3, Step 3 can be implemented more efficiently. Specifically, we can obtain $\hat{w}_t$ by performing a topological sort of the directed acyclic graph $([d], A_t)$ and assigning values to $\hat{w}_t$ according to the obtained order. This ensures $\hat{w}_t(i) > \hat{w}_t(j)$ for all $(i,j) \in A_t$, and the time complexity of this procedure is $O(d + |A_t|) = O(d^2)$. Further details on the approximate center-of-gravity implementation are given in Appendix A, per-round computational costs under standard oracle models are discussed in Appendix B, and a synthetic experiment is reported in Appendix C.

## 5. $O((C+1)d \log d)$-Regret Algorithm under Corruptions

We extend Algorithm 1 to handle corrupted instances, where the agent's actions are adversarially corrupted in up to $C$ rounds. We show that a simple modification yields a regret upper bound of $O((C+1)d \log d)$, without using any prior knowledge of $C$. Our high-level idea is to restart Algorithm 1 whenever we detect a corruption that prevents us from guaranteeing that we incur no regret. To this end, we utilize the acyclicity of the directed graph formed by $A_t$.

Recall that $A_t$ maintains a set of pairs of indices $(i,j)$ such that $w^*(i) > w^*(j)$ must hold for $x_1, \ldots, x_{t-1}$ to be optimal for $w^*$. Consider the directed graph $([d], A_t)$. A key observation is that if $([d], A_{t+1})$ has a cycle, at least one of the observed actions is suboptimal, i.e., corrupted.[7] For later use, we state this in a general form for any interval of rounds.

---

[7]This implication uses the distinctness condition in Assumption 2.2. If tied components are allowed, a directed cycle may instead indicate that the corresponding coordinates of $w^*$ have equal values, and we cannot distinguish such cycles from those caused by corruptions. In the uncorrupted setting, this can be handled by contracting strongly connected components, as in Appendix D.

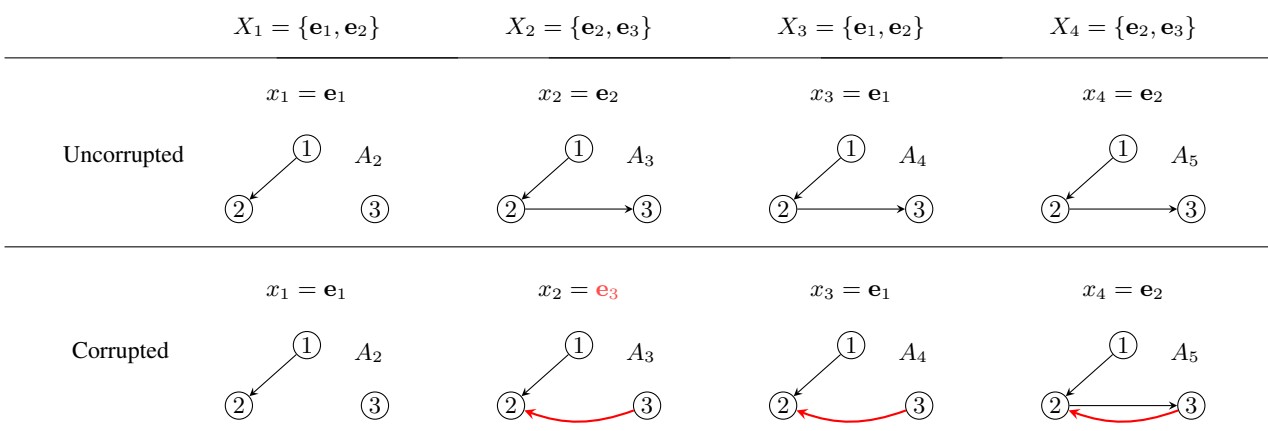

*Figure 2.* Examples of directed graphs $([d], A_{t+1})$ in uncorrupted and corrupted sequences; note that the graph for $A_1 = \emptyset$ is omitted. Let $d = 3$ and $w^* \in \mathbb{R}^3$ satisfy $w^*(1) > w^*(2) > w^*(3)$. Feasible sets $X_t$ are given in the top row. In the uncorrupted case, $A_{t+1}$ remains acyclic for all $t$. In the corrupted case, $x_2$ (shown in red) is corrupted, adding a wrong arc $3 \to 2$ to $A_3$. Still, $A_3$ remains acyclic and does not yet indicate the corruption. This does not affect the choice from $X_3 = \{\mathbf{e}_1, \mathbf{e}_2\}$ in the next round, and we correctly select $\hat{x}_3 = x_3 = \mathbf{e}_1$. Later, when $x_4$ is observed, the correct arc $2 \to 3$ is added, creating a cycle $2 \to 3 \to 2$ in $A_5$, revealing the corruption.

**Lemma 5.1.** *Let $t', t \in \{0, \dots, T\}$ with $t' < t$ and define the arc set for this interval as*

$$A_{t':t+1} = \bigcup_{s=t'+1}^{t} \{ (i,j) : i \neq j, \ x_s - \mathbf{e}_i + \mathbf{e}_j \in X_s \}.$$

*If $([d], A_{t':t+1})$ has a directed cycle, then at least one of $x_{t'+1}, \dots, x_t$ is suboptimal.*

*Proof.* Assume for contradiction that for $s = t' + 1, \dots, t$, $x_s \in \arg\max_{x \in X_s} \langle w^*, x \rangle$ holds. Take any directed cycle $i_1 \to i_2 \to \cdots \to i_{k+1} = i_1$ of $([d], A_{t':t+1})$. By the definition of $A_{t':t+1}$, for each arc $(i_l, i_{l+1})$ in the cycle, there exists $s_l \in \{t'+1, \dots, t\}$ with $x_{s_l} - \mathbf{e}_{i_l} + \mathbf{e}_{i_{l+1}} \in X_{s_l}$. Since $x_{s_l}$ is optimal for $w^*$ and the components of $w^*$ are distinct, $w^*(i_l) > w^*(i_{l+1})$ holds for $l = 1, \dots, k$, hence

$$w^*(i_1) > w^*(i_2) > \cdots > w^*(i_k) > w^*(i_1),$$

which is a contradiction. $\square$

The lemma suggests that monitoring the acyclicity of $([d], A_{t':t+1})$ provides useful information for detecting corruption. The inverse implication, however, is not necessarily true: the acyclicity of $([d], A_{t':t+1})$ does not imply the optimality of $x_{t'+1}, \dots, x_t$. Still, if $([d], A_{t':t})$ is acyclic, we can take $\hat{w}_t$ required in Step 3 of Algorithm 1, such that $\hat{w}_t(i) > \hat{w}_t(j)$ for all $(i, j) \in A_{t':t}$. Then, similar to the proof of Theorem 3.1, if $A_{t':t+1} = A_{t':t}$, we can guarantee that playing $\hat{x}_t \in \arg\max_{x \in X_t} \langle \hat{w}_t, x \rangle$ yields no regret at round $t$, even in the presence of corruptions before round $t$. The proof is essentially the same as that of Theorem 3.1, but we explicitly present it here to highlight that the claim still holds under potential corruptions.

**Lemma 5.2.** *Let $t', t \in \{0, \dots, T\}$ with $t' < t$. Assume that the following conditions hold:*

1. *$([d], A_{t':t})$ is acyclic, and*

2. *$A_{t':t} = A_{t':t+1}$.*

*Then, by taking any $\hat{w}_t$ with distinct components such that $\hat{w}_t(i) > \hat{w}_t(j)$ for all $(i, j) \in A_{t':t}$, we have $\hat{x}_t = x_t$.*

*Proof.* By the definition of $A_{t':t+1}$, $\hat{w}_t(i) > \hat{w}_t(j)$ holds for all $(i, j)$ with $i \neq j$ and $x_t - \mathbf{e}_i + \mathbf{e}_j \in X_t$. Therefore, Proposition 2.4 ensures $x_t \in \arg\max\{\langle \hat{w}_t, x \rangle : x \in X_t\}$, and the maximizer set is a singleton, $\{\hat{x}_t\}$, as components of $\hat{w}_t$ are distinct, hence $\hat{x}_t = x_t$. $\square$

Figure 2 illustrates examples of directed graphs $([d], A_{t+1})$ for uncorrupted and corrupted sequences. In the corrupted sequence, the corruption at $t = 2$ does not immediately create a cycle in $A_3$, but it is eventually revealed as a cycle at $t = 4$, consistent with Lemma 5.1. At $t = 3$, because $A_3 = A_4$, we can play $\hat{x}_3 = x_3$ and incur no regret despite the earlier corruption, as ensured by Lemma 5.2.

Thanks to the above two lemmas, we can effectively handle corruptions by monitoring the acyclicity of $([d], A_t)$: if $([d], A_{t':t+1})$ turns out to have a cycle, then there must be at least one corrupted round in $\{t'+1, \dots, t\}$ by Lemma 5.1; if $([d], A_{t':t})$ is acyclic and $A_{t':t} = A_{t':t+1}$ holds, we can play $\hat{x}_t$ with no regret at round $t$ by Lemma 5.2. Therefore, by running Algorithm 1 and restarting it whenever $([d], A_{t+1})$ turns out to have a cycle in Step 7, we can achieve a regret bound of $O((C+1)d \log d)$ without knowing $C$ in advance. We present the resulting procedure in Algorithm 2.

**Algorithm 2** Algorithm for corrupted feedback

1: Let $A_1 = \emptyset$.
2: **for** $t = 1, 2, \dots, T$ **:**
3:     Take $\hat{w}_t$ with distinct components such that
          $\hat{w}_t(i) > \hat{w}_t(j)$    for all $(i, j) \in A_t$,
       obtained by taking the center of gravity of $P_t$ and
       breaking tied components, if any, via an arbitrarily
       small perturbation within $P_t$.
4:     Let $\hat{x}_t \in \arg\max\{\langle \hat{w}_t, x \rangle : x \in X_t\}$.
5:     Observe the agent's action $x_t$.
6:     $A_{t+1} \leftarrow A_t \cup \{(i, j) : i \neq j, x_t - \mathbf{e}_i + \mathbf{e}_j \in X_t\}$.
7:     Restart: if $([d], A_{t+1})$ has a cycle, set $A_{t+1} = \emptyset$.

**Theorem 5.3.** *Algorithm 2 achieves*

$$R_T = O((C+1)d \log d).$$

*Proof.* Let $t'$ denote a round such that $([d], A_{t'+1})$ turns out to have a cycle in Step 7 and the algorithm is restarted. Let $t > t'$ be the next round such that $([d], A_{t+1})$ has a cycle in Step 7. Within $s = t' + 1, \dots, t$, $([d], A_{t':s})$ remains acyclic. If $A_{t':s} = A_{t':s+1}$, Lemma 5.2 ensures $\hat{x}_s = x_s$, yielding no regret. Otherwise, $\hat{x}_s \neq x_s$ may happen; in this case, Lemma 4.1, which is proved without using optimality of $x_s$ for $w^*$, implies that the volume of $P_s$ (defined for $s \in \{t' + 1, \dots, t\}$, starting from $P_{t'+1} = [0, 1]^d$) decreases by a factor of $1 - 1/e$. As long as $([d], A_{t':s})$ is acyclic, $P_s$ contains at least one order simplex, hence $\mathrm{Vol}(P_{t'+1})/\mathrm{Vol}(P_t) \leq d!$. Therefore, by the same proof as that of Theorem 4.2, the regret accumulated for $s = t' + 1, \dots, t$ is $O(d \log d)$.

Because $([d], A_{t+1})$ has a cycle, Lemma 5.1 implies that at least one of $x_{t'+1}, \dots, x_t$ is corrupted, triggering a restart. Therefore, the algorithm is restarted at most $C$ times over $T$ rounds, and the regret accumulated for each interval between two consecutive restarts is $O(d \log d)$, as discussed above. Consequently, the regret over $T$ rounds is $O((C + 1)d \log d)$, completing the proof. $\square$

**Time complexity.** The additional time complexity incurred by the corruption-robust modification is due to cycle detection in Step 7. We can use the standard depth-first search or topological sort for this purpose, taking $O(d^2)$ time. Thus, the per-round time complexity of the corruption-robust algorithm increases by $O(d^2)$ compared to the uncorrupted case, which is negligible compared to the other parts as discussed at the end of Section 4.

## 6. Lower Bound

To complement our upper bounds, we present a regret lower bound of $\Omega(d)$ that holds even for the M-convex setting. The proof is mostly based on that of Sakaue et al. (2025b, Theorem 5.1), but we take care of the M-convexity of action sets.

**Theorem 6.1.** *There exists an instance of online inverse linear optimization such that $X_1, \dots, X_T$ are M-convex sets and any randomized algorithm incurs a regret of*

$$\mathbb{E}[R_T] = \Omega(d).$$

*Proof.* Sakaue et al. (2025b, Theorem 5.1) obtain an $\Omega(m)$ lower bound in dimension $m$ by using instances such that each feasible set is an axis-aligned line segment: for some $i_t \in [m]$, the coordinate $x(i_t)$ ranges over an interval and all other coordinates are fixed to zero. Take $m = (4k)^2$ for some $k \in \mathbb{Z}_{>0}$, so that $\sqrt{m}/4 = k$ is an integer. The same lower-bound argument applies after restricting each line segment to its integer points, because the forward optimization problem is linear and the maximum over a line segment is attained at an endpoint. Write the resulting integer feasible set as $Y_t \subseteq \mathbb{Z}^m$.

The set $Y_t$ is not M-convex as a subset of $\mathbb{Z}^m$, but it is $\mathrm{M}^\natural$-convex, and the standard transformation from $\mathrm{M}^\natural$-convex sets to M-convex sets applies (Murota, 2003, Sections 4.7 and 6.1). Define an embedded set in $\mathbb{Z}^{m+1}$ by

$$X_t' = \left\{ x' \in \mathbb{Z}^{m+1} : \begin{array}{l} (x'(1), \dots, x'(m)) \in Y_t, \\ x'(0) = -\sum_{i=1}^m x'(i) \end{array} \right\}.$$

This $X_t'$ is M-convex: the additional coordinate records the negative total mass of the original coordinates, and hence all vectors in $X_t'$ have the same coordinate sum.

It remains to check that this embedding preserves the inverse optimization instance. For an objective vector $w \in \mathbb{R}^m$ in the original lower-bound instance, define $w' \in \mathbb{R}^{m+1}$ by $w'(0) = 0$ and $w'(i) = w(i)$ for $i \in [m]$. Then, for every $x' \in X_t'$ corresponding to $y = (x'(1), \dots, x'(m)) \in Y_t$, we have $\langle w', x' \rangle = \langle w, y \rangle$. Thus, the embedded instance has exactly the same optimal actions, after ignoring the dummy coordinate, and the same regret values as the original lower-bound instance. Consequently, any algorithm for M-convex action sets in dimension $m + 1$ would yield an algorithm for the original hard instance in dimension $m$ with the same regret; therefore, the $\Omega(m)$ lower bound transfers. This gives an $\Omega(d)$ lower bound after renaming the dimension as $d$. $\square$

## 7. Conclusion and Discussion

We have studied online inverse linear optimization with M-convex action sets and established finite regret bounds that scale polynomially in the dimension: $O(d \log d)$ for uncorrupted feedback, and $O((C + 1)d \log d)$ under up to $C$ adversarial corruptions without knowing $C$ in advance. Our analysis combines a structural characterization of optimal solutions on M-convex sets with the geometric volume argument, and it extends to the corruption-robust setting via monitoring of directed graphs constructed from feedback.

The results have partially closed the open question in the prior literature, and we have shown that broad classes of combinatorial action sets admit the improved regret bounds.

Several directions remain open for future research. Closing the remaining $O(\log d)$ gap between the upper and lower bounds, and exploring the tight dependence on the number of corruptions are intriguing challenges. Establishing corruption-robust bounds that depend on the cumulative suboptimality, as considered in Sakaue et al. (2025b) and Sakaue (2026), is another interesting direction. Another potential direction is to connect our setting with reward estimation and inverse reinforcement learning for bandit models (Guha et al., 2024; Petriconi and Vernade, 2024). Our regret criterion evaluates the quality of the actions induced by the learner's estimates, rather than the estimation error of $w^*$ itself. Moreover, our analysis relies on observing (approximate) optimizers of the forward problem, whereas in inverse bandit settings the observed actions may be generated by an online learning algorithm. Extending our M-convexity-based approach to such settings would require additional ideas. More broadly, as illustrated in Figure 2, M-convex action sets subsume two-action settings, which can be viewed as a simple model of preference feedback—a burgeoning topic in human-in-the-loop machine learning (Christiano et al., 2017; Ouyang et al., 2022). In such scenarios, the agent's actions and objectives can exhibit complex structure, typically represented in high-dimensional spaces. On the other hand, the lower bound of $\Omega(d)$ arises when the learner must infer each component of $w^*$ separately. In this light, understanding what structures of the ambient space of $w^*$ and action sets $X_t$ permit faster learning beyond the $\Omega(d)$ barrier is an important direction for future work.

## Acknowledgements

Taihei Oki was supported by JST FOREST Grant Number JPMJFR232L, JSPS KAKENHI Grant Number JP22K17853, and JST BOOST Program Grant Number JPMJBY25A6. Shinsaku Sakaue was supported by JST BOOST Program Grant Number JPMJBY24D1. The authors thank the anonymous reviewers for their helpful reviews and comments.

## Impact Statement

This paper presents work whose goal is to advance the field of Machine Learning. There are many potential societal consequences of our work, none of which we feel must be specifically highlighted here.

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

# A. Approximate Center of Gravity

We provide a polynomial-time implementation of the center-of-gravity computation required in Section 4. Recall

$$P_t = \big\{ w \in [0,1]^d \ : \ w(i) \geq w(j) \text{ for all } (i,j) \in A_t \big\}.$$

We assume that $([d], A_t)$ is acyclic, as is the case in the uncorrupted setting and within each interval between two consecutive restarts in the corrupted setting, with $A_t$ interpreted as the interval-local arc set.

**Membership oracle and an interior point.** A membership query to $P_t$ can be answered by checking the box constraints and the order constraints induced by $A_t$, which takes $O(d^2)$ time. To initialize the random walk, take a topological ordering $\pi$ of $([d], A_t)$ and define $q(\pi(k)) = 1 - \frac{k}{d+1}$ for $k = 1, \ldots, d$. This definition ensures that $q$ belongs to $P_t$. Moreover, for every order constraint $w(i) \geq w(j)$ defining $P_t$, the Euclidean distance from $q$ to the hyperplane $w(i) = w(j)$ is at least $1/((d+1)\sqrt{2})$, and the distance from $q$ to the boundary of $[0,1]^d$ is at least $1/(d+1)$. Therefore, we have

$$R_0 \mathbb{B}^d \subseteq P_t - q \subseteq R_1 \mathbb{B}^d \quad \text{with} \quad R_0 = \frac{1}{(d+1)\sqrt{2}} \quad \text{and} \quad R_1 = \sqrt{d}.$$

In particular, the ratio $R_1/R_0$ is $O(d^{3/2})$.

Let $z(P_t)$ denote the center of gravity of $P_t$. By applying the hit-and-run algorithm of Lovász and Vempala (2006) through the guarantee of Feldman et al. (2015, Theorem 5.7), for any $\tau > 0$ and $\delta > 0$, with probability at least $1 - \delta$, we can compute a point $\hat{w}_t \in P_t$ that satisfies

$$\|\hat{w}_t - z(P_t)\|_{E_{P_t}} \leq \tau,$$

where $\|\cdot\|_{E_{P_t}}$ denotes the norm induced by the inertial ellipsoid of $P_t$. The $\tilde{O}$ notation below follows Feldman et al. (2015, Theorem 5.7): after substituting $R_1/R_0 = O(d^{3/2})$, we keep the factors corresponding to $\log(R_1/R_0)$ and $\log(1/\delta)$ explicit, while suppressing the additional polylogarithmic factors from the random-walk implementation. The number of membership-oracle calls is $\tilde{O}\big(d^4 \log d \cdot \frac{\log(1/\delta)}{\tau^2}\big)$. Since each membership query takes $O(d^2)$ time, the time required for this approximate center-of-gravity computation per round is

$$\tilde{O}\bigg(d^6 \log d \cdot \frac{\log(1/\delta)}{\tau^2}\bigg).$$

**Volume decrease.** We next check that replacing the exact center of gravity by the approximate one preserves the regret analysis up to constant factors. Suppose that $x_t \neq \hat{x}_t$ holds. By the same argument as in the proof of Lemma 4.1, there exists $(i,j) \in A_{t+1} \setminus A_t$ such that $\hat{w}_t(i) < \hat{w}_t(j)$. Define the halfspace $H$, as in the proof of Lemma 4.1, by

$$H = \big\{ w \in \mathbb{R}^d \ : \ w(i) \geq w(j) \big\}.$$

Since $\hat{w}_t(i) < \hat{w}_t(j)$, the complementary halfspace contains $\hat{w}_t$. The approximate Grünbaum theorem of Feldman et al. (2015, Lemma 5.6) implies

$$\text{Vol}(P_t \setminus H) \geq \bigg(\frac{1}{e} - \tau\bigg)\text{Vol}(P_t).$$

Here, using the open complement instead of the closed complementary halfspace does not change the volume because the boundary hyperplane has zero Lebesgue measure. Since $P_{t+1} \subseteq P_t \cap H$, we obtain

$$\text{Vol}(P_{t+1}) \leq \text{Vol}(P_t \cap H) = \text{Vol}(P_t) - \text{Vol}(P_t \setminus H) \leq \bigg(1 - \frac{1}{e} + \tau\bigg)\text{Vol}(P_t).$$

Thus, for any constant $\tau < 1/e$, each mistake decreases the volume by a constant factor.

**Regret bound.** For example, set $\tau = \frac{1}{2e}$. Taking $\delta = 1/T^2$ for each round and applying a union bound, all approximate center-of-gravity computations succeed with probability at least $1 - 1/T$. Conditioned on this event, the proof of Theorem 4.2 gives an upper bound of

$$\log_{(1-1/e+\tau)^{-1}} d! = O(d \log d)$$

on the number of mistake rounds. On the complement event, the regret is $O(T)$ by Assumption 2.2, and hence its contribution to the expected regret is $O(1)$. Consequently, the randomized implementation based on approximate centers of gravity satisfies $\mathbb{E}[R_T] = O(d \log d)$. The same argument applies to Algorithm 2 within each interval between two consecutive restarts, yielding $\mathbb{E}[R_T] = O((C+1)d \log d)$.

*Table 2.* Regret and running time on time-varying uniform matroids. Regret intervals are 95% confidence intervals over five independent trials.

| Algorithm | Regret | Time per round |
|---|---|---|
| Algorithm 1, $O(d^2)$ version | $1.640 \pm 0.339$ | $11\,\mu\mathrm{s}$ |
| Algorithm 1, $O(d\log d)$ version ($N = 10^2$) | $1.677 \pm 0.260$ | $23\,\mu\mathrm{s}$ |
| Algorithm 1, $O(d\log d)$ version ($N = 10^3$) | $1.491 \pm 0.156$ | $120\,\mu\mathrm{s}$ |
| Algorithm 1, $O(d\log d)$ version ($N = 10^4$) | $1.201 \pm 0.382$ | $1.1\,\mathrm{ms}$ |
| ONS (Sakaue et al., 2025b) | $3.510 \pm 0.776$ | $70\,\mu\mathrm{s}$ |
| OGD (Bärmann et al., 2020) | $4.848 \pm 1.105$ | $4\,\mu\mathrm{s}$ |

## B. Detailed Computational Costs

We detail the per-round computational costs of the algorithms under standard oracle access.

**General M-convex sets.** Let $C_{\mathrm{mem}}$ denote the time required for one membership-oracle query to $X_t$, and let $C_{\mathrm{lin}}$ denote the time required for one linear optimization problem over the M-convex set $X_t$. The latter can be implemented in polynomial time given a membership oracle for $X_t$ by the algorithms of Shioura and Tanaka (2007, Theorems 3.1 and 3.2), since M-convex sets are special cases of jump systems. For the $O(d^2)$-regret implementation of Algorithm 1, Step 3 takes $O(d^2)$ time by topological sorting. Step 4 takes $C_{\mathrm{lin}}$ time, and Step 6 takes $O(d^2 C_{\mathrm{mem}})$ time by checking all $O(d^2)$ pairs $(i, j)$ and querying whether $x_t - \mathbf{e}_i + \mathbf{e}_j \in X_t$. Thus, the per-round time of this implementation is $C_{\mathrm{lin}} + O(d^2 C_{\mathrm{mem}} + d^2)$. For the $O(d\log d)$-regret implementation using the approximate center of gravity from Appendix A, Step 3 instead takes $\tilde{O}(d^6 \log d \cdot \log T)$ time per round, where we take $\tau$ as a constant and $\delta = 1/T^2$. Therefore, the per-round time is

$$\tilde{O}(d^6 \log d \cdot \log T) + O\big(C_{\mathrm{lin}} + d^2 C_{\mathrm{mem}} + d^2\big).$$

For Algorithm 2, the same per-round bounds apply, with the $O(d^2)$ cycle-detection time included in the $d^2$ term.

**Matroid bases.** Consider the important special case where $X_t$ is the set of characteristic vectors of bases of a matroid over $[d]$, and suppose that an independence oracle is available. Let $C_{\mathrm{ind}}$ denote the time required for one independence-oracle query. This instantiates the preceding bounds with $C_{\mathrm{mem}} = C_{\mathrm{ind}}$ and $C_{\mathrm{lin}} = O(d\log d + dC_{\mathrm{ind}})$ (up to lower-order bookkeeping absorbed in the $d^2$ term). Indeed, linear optimization over matroid bases is solved by the standard greedy algorithm: sort the elements by $\hat{w}_t$ and add each element whenever doing so preserves independence. Moreover, in Step 6, feasibility of $x_t - \mathbf{e}_i + \mathbf{e}_j$ can be tested by one independence-oracle query for each pair $(i, j)$. For the $O(d^2)$-regret implementation of Algorithm 1, Step 3 is a topological sort of $([d], A_t)$ and hence takes $O(d^2)$ time. Therefore, in the matroid case its per-round time is dominated by Step 6, which is $O(d^2 C_{\mathrm{ind}})$. For the $O(d\log d)$-regret implementation using the approximate center of gravity from Appendix A, the per-round time is

$$\tilde{O}(d^6 \log d \cdot \log T) + O(d^2 C_{\mathrm{ind}}),$$

where we take $\tau$ as a constant and $\delta = 1/T^2$. For Algorithm 2, the same per-round bounds apply, with an additional $O(d^2)$ time for cycle detection.

## C. Numerical Experiment

We provide a synthetic experiment to illustrate the effect of exploiting the M-convex structure. The code is available at https://github.com/ssakaue/online-inverse-m-convex-public. The feasible sets are time-varying uniform matroids. There are $d = 20$ items, and in each round $k = 10$ items are unavailable. Given the remaining $d - k = 10$ available items, the feasible set $X_t$ consists of all subsets of size $m = 5$. We set $T = 5000$ and report the average over five independent trials. For Algorithm 1 with the $O(d\log d)$-regret implementation, we approximate the center of gravity by $N$ hit-and-run iterations. We compare our two algorithms with ONS (Sakaue et al., 2025b) and OGD (Bärmann et al., 2020).

The results in Table 2 show that our algorithms specialized to M-convex sets achieve smaller regret than the two baselines in this experiment. The performance of the $O(d\log d)$-regret implementation improves as the number of hit-and-run iterations increases, which is consistent with the role of the center-of-gravity approximation in the volume argument.

## D. On Handling Tied Components

Assumption 2.2 assumes that the components of $w^*$ are distinct. This assumption simplifies the presentation, especially the use of acyclicity of $([d], A_t)$. Here, we clarify how ties can be treated in the uncorrupted setting, and why the distinction matters in the corrupted setting.

Suppose first that the feedback is uncorrupted. For any arc $(i, j) \in A_t$, optimality of the observed action and Proposition 2.4 imply $w^*(i) \geq w^*(j)$. Thus, if $i$ and $j$ belong to the same strongly connected component of $([d], A_t)$, then both $w^*(i) \geq w^*(j)$ and $w^*(j) \geq w^*(i)$ hold, and hence $w^*(i) = w^*(j)$. Therefore, the vertices in each strongly connected component can be contracted without changing the objective values relevant to regret. After this contraction, the condensation graph is acyclic. One can then choose the estimate to be constant on each strongly connected component and strictly ordered along the condensation graph. Any ambiguity in the selected maximizer is confined to coordinates in the same component, on which $w^*$ has equal values, and hence does not contribute to regret. The volume argument in Section 4 is applied in the quotient space of the contracted components, whose dimension is at most $d$. Thus, the same arguments as in Sections 3 and 4 apply to the contracted components.

In contrast, in the corrupted case, the distinctness assumption is used to interpret a directed cycle as a certificate of corruption. Indeed, the proof of Lemma 5.1 derives a contradiction from $w^*(i_1) > w^*(i_2) > \cdots > w^*(i_k) > w^*(i_1)$. If ties are allowed, a cycle may instead indicate that the corresponding coordinates have equal objective values. For this reason, the present corruption-detection mechanism relies on the distinctness condition, whereas the uncorrupted regret argument can be formulated after contracting strongly connected components.

