# OpenReview forum: "Finite and Corruption-Robust Regret Bounds in Online Inverse Linear Optimization under M-Convex Action Sets"
_ICML.cc/2026/Conference — ICML 2026 regular_

### Official Review · Reviewer_KXCk · 2026-02-24

**Soundness:** 3
**Presentation:** 3
**Significance:** 3
**Originality:** 3
**Overall Recommendation:** 5
**Confidence:** 3

**Summary:**

This paper studies online inverse linear optimization, also called contextual recommendation, where the learner observes an agent's actions over changing feasible sets and tries to minimize regret with respect to the agent's unknown linear objective. The main result is that when the feasible action sets are M-convex, a broad discrete-convex family that includes matroids, one can achieve a finite regret bound of $O(d \log d)$, partially answering the open question of whether finite regret polynomial in $d$ is possible. The analysis extracts pairwise order constraints from optimal actions on M-convex sets and combines them with a center-of-gravity argument plus Grunbaum-style volume shrinkage to control the number of mistakes. The paper also handles adversarial corruption in up to $C$ rounds, obtaining $O((C+1)d \log d)$ regret without knowing $C$ in advance by monitoring cycles in a directed graph of inferred order relations and restarting when a cycle appears. An $\Omega(d)$ lower bound for the M-convex setting shows the rate is near-tight up to the logarithmic factor.

**Compliance With Llm Reviewing Policy:**

Affirmed.

**Final Justification:**

I would like to recommend an acceptance.

**Key Questions For Authors:**

1. Practical implementability: can you give explicit guarantees when the learner uses only an approximate center of gravity, for example via random-walk sampling? How does that approximation feed into the volume-shrinkage step and the final regret bound?
2. Concrete complexity for matroids: under standard matroid oracles, what clean per-round runtime guarantee can you provide? In particular, can the exchange-implied constraint update be implemented without an expensive enumeration of many pairs?
3. Robustness model: do you see a route from the current "at most $C$ corrupted rounds" analysis to a cumulative suboptimality notion with many small corruptions, while still keeping a finite regret guarantee with no $\log T$ term?

**Limitations:**

No. The paper would benefit from a more explicit discussion of the practical cost of approximate center-of-gravity computation, the narrowness of the corruption model, and why broader societal risk is limited in this mainly theoretical setting.

**Strengths And Weaknesses:**

**Strengths**

What stood out most to me is that the paper makes real theoretical progress on finite regret in a structured combinatorial setting, rather than just shaving constants off existing logarithmic-in-$T$ results. M-convexity is not cosmetic here; it is the structural reason observed optimal actions can be converted into useful order information. I also thought the corruption-robust extension fit naturally with the main framework.
-The headline guarantee of finite regret $O(d \log d)$ under M-convex action sets seems genuinely important relative to prior finite bounds that were exponential in $d$ or still retained $\log T$ dependence.
- The proof strategy is elegant. The exchange structure of M-convex sets combines cleanly with the center-of-gravity and volume-shrinkage argument, and the overall mistake-bound picture is easy to follow.
- The corruption-handling mechanism is appealingly simple: restart when the inferred order graph develops a cycle. It does not need prior knowledge of $C$ and still yields the stated $O((C+1)d \log d)$ regret bound.
- The $\Omega(d)$ lower bound is useful because it clarifies that the remaining gap is only logarithmic, rather than leaving the tightness story vague.

**Weaknesses**

My biggest reservation is about implementability. As often happens with center-of-gravity arguments, the theory is attractive but the computational model underneath it is not fully transparent. I also think the paper could do a better job stating exactly how dependent the phenomenon is on M-convexity and on the chosen corruption model.
- The paper already notes that exact center-of-gravity computation is intractable and points to approximate alternatives, but it remains unclear how those approximations quantitatively affect the volume-shrinkage argument and the final regret guarantee.
- The oracle model for M-convex sets deserves a more explicit discussion. For important subclasses like matroids, a concrete per-round runtime guarantee under standard oracle access would make the contribution easier to assess operationally.
- The robust setting counts the number of corrupted rounds, rather than cumulative deviation or small-but-frequent suboptimality. I am unsure whether the approach extends to that more graded notion while retaining finite regret independent of $\log T$, and I would like the paper to discuss that boundary more directly.

---

> ### Author Rebuttal · Authors · 2026-03-26
>
> We thank the reviewer for the thorough and positive review. We address the main questions below.
>
> ## **Q1 / W1. Guarantees with approximate center of gravity.**
>
> Please see our response to Reviewer ef6R (W2) for the detailed analysis. Here we directly address the three sub-questions.
>
> **Approximation guarantee.** Using the hit-and-run algorithm of Lovász and Vempala (2006), Feldman et al. (2015, Theorem 5.7) guarantee that for any target accuracy $\tau > 0$ and failure probability $\delta > 0$, with probability $\ge 1-\delta$ one can compute $\hat{w}\_t$ satisfying $\\|\hat{w}\_t - z(P\_t)\\|\_{E\_{P\_t}} \le \tau$ in $\tilde{O}(d^4 \log d \cdot \log(1/\delta)/\tau^2)$ membership-oracle calls, where $\\|\cdot\\|\_{E\_{P\_t}}$ is the inertial ellipsoid norm and $z(P\_t)$ is the exact centroid. Each oracle call takes $O(d^2)$ time (checking $O(d^2)$ box and order constraints).
>
> **Volume shrinkage.** The approximate Grünbaum's theorem (Feldman et al. 2015, Lemma 5.6) gives $\mathrm{Vol}(P\_t \setminus H\_{ij}) \ge (1/e - \tau)\mathrm{Vol}(P\_t)$, where $H\_{ij} = \\{w : w(i) \ge w(j)\\}$. Since $P\_{t+1} \subseteq P\_t \cap H\_{ij}$, we get $\mathrm{Vol}(P\_{t+1}) \le (1 - 1/e + \tau)\mathrm{Vol}(P\_t)$.
>
> **Final regret bound.** For any constant $\tau < 1/e$ (e.g., $\tau = 1/(2e)$), the volume shrinks by a constant factor per mistake, and the argument of Theorem 4.2 carries through unchanged. Taking the union bound over all $T$ rounds with $\delta = 1/T^2$ leads to $\mathbb{E}[R\_T] = O(d\log d)$. This approximate centroid computation runs in $\tilde{O}(d^6 \log d \cdot \log T)$ time per round. We will add this analysis to the revision.
>
> ## **Q2 / W2. Concrete complexity for matroids.**
>
> When action sets $X_t$ are matroid bases with the standard independence oracle (given $S \subseteq [d]$, return whether $S$ is independent), the major computational steps of Algorithm 2 are analyzed as follows:
>
> - **Step 3** (computing $\hat{w}_t$ as the approximate center of gravity of $P_t$): Using the hit-and-run algorithm of Lovász and Vempala (2006) via Feldman et al. (2015, Theorem 5.7) with constant $\tau < 1/e$ and $\delta = 1/T^2$, this takes $\tilde{O}(d^6 \log d \cdot \log T)$ time per round (as in our response to Q1/W1 above).
> - **Step 4** (linear optimization over matroid bases): The classic greedy algorithm—sort elements by $\hat{w}_t$, greedily add elements maintaining independence—runs in $O(d\log d + d C\_{\text{ind}})$ time, where $C\_{\text{ind}}$ is the time complexity of a single independence oracle query.
> - **Step 6** (updating $A\_{t+1}$): Check all $O(d^2)$ pairs $(i,j)$ whether $x_t - e_i + e_j$ is a feasible basis, using one independence oracle query per pair. This takes $O(d^2 C\_{\text{ind}})$ time.
> - **Step 7** (cycle detection): This can be done in $O(d^2)$ time using depth-first search.
>
> Thus, the per-round time complexity for matroids is dominated by the centroid computation in Step 3 and the update of $A\_{t+1}$ in Step 6, giving $\tilde{O}(d^6 \log d \cdot \log T + d^2 C\_{\text{ind}})$ time per round.
>
> For the $O(d^2)$-regret algorithm (Algorithm 1), Step 3 is replaced by a simple update of $\hat{w}_t$ that runs in $O(d^2)$ time. Consequently, its per-round time complexity is dominated by Step 6, giving $O(d^2 C\_{\text{ind}})$ time per round.
>
> ## **Q3 / W3. Extension to cumulative suboptimality corruption model.**
>
> This is an important open direction, as noted in Sections 2.2 and 7. Our cycle-detection mechanism provides a definitive witness of corruption: a directed cycle in $([d], A_t)$ certifies that at least one observed action is suboptimal. However, with many small corruptions (small cumulative suboptimality, large $C$), the graph may develop cycles repeatedly, causing many restarts and degrading the bound. Handling the suboptimality-based corruption model within this framework alone currently seems challenging.
>
> Among the existing results summarized in Table 1, Sakaue et al. (2025) is the only one that handles the corruption model depending on the cumulative suboptimality $\Delta_T$. Their analysis relies on an online-convex-optimization technique, specifically MetaGrad (van Erven and Koolen, 2016), which fundamentally differs from our cycle-detection approach. In our current view, a promising route is to integrate our M-convexity-based arguments into the framework of Sakaue et al. (2025) to make their regret bound finite while retaining the $\Delta_T$ dependence.

---

> > ### Author Rebuttal · Reviewer_KXCk · 2026-04-01
> >
> > I appreciate the authors' effort in providing a thorough rebuttal. I plan to keep my original ratings. Thanks.

---

### Official Review · Reviewer_1zLJ · 2026-03-10

**Soundness:** 3
**Presentation:** 2
**Significance:** 2
**Originality:** 3
**Overall Recommendation:** 4
**Confidence:** 4

**Summary:**

This paper studies online inverse linear optimization under an M-convex assumption on the feasible sets. The authors first present a simple algorithm with regret $O(d^2)$, based on learning the pairwise order relations implied by M-convexity. They then improve this to $O(d\log d)$ by selecting the estimate as the center of gravity of the consistent region. To the best of my understanding, this gives the first finite regret bounds that are polynomial in $d$ for this inverse optimization setting under this structural assumption. The paper further extends the method to the case of up to $C$ adversarial corruptions, obtaining a regret bound of $O((C+1)d\log d)$ without requiring prior knowledge of $C$. Finally, it provides an $\Omega(d)$ lower bound for the M-convex case, showing near-tightness up to a logarithmic factor.

**Compliance With Llm Reviewing Policy:**

Affirmed.

**Final Justification:**

My primary remaining concern is that the assumption of M-convexity is quite strong, which I believe limits the scope and significance of the results. For this reason, my assessment of the significance is fair, and my overall recommendation is weak accept.

**Key Questions For Authors:**

Please address the weaknesses above. I also have the following specific questions:

* **Q1:** The analysis assumes that the components of $w^* $ are distinct. Could the authors clarify more explicitly what changes if some coordinates of $w^* $ are equal?

* **Q2:** In Algorithm 1, could the authors explain more clearly why Step 4 can be solved in polynomial time using only a membership oracle for $X_t$?

**Limitations:**

yes

**Strengths And Weaknesses:**

## Strengths

* This is a solid theory paper that makes progress on an open question in online inverse optimization, namely whether one can obtain a finite regret bound polynomial in the dimension. Under the M-convex assumption, the paper provides a positive answer.
* The paper uses M-convexity in a meaningful and technically natural way. The approach relies on the exchange-based characterization of optimality in Proposition 2.4, and this structure is exactly what enables both the $O(d^2)$ bound and the sharper $O(d\log d)$ result.
* The corruption extension is meaningful and fits the main idea of the paper well. The restart rule based on cycle detection is natural, and the resulting $O((C+1)d\log d)$ guarantee is a valuable addition.

----

## Weaknesses

* **W1:** The M-convex assumption is quite strong, and it is precisely this structure that makes the problem tractable via Proposition 2.4. As a result, the paper should be viewed as establishing a positive result in a structured setting, rather than providing a general resolution of whether inverse optimization admits finite regret polynomial in the dimension.
* **W2:** The $O(d\log d)$ algorithm relies on computing the center of gravity, but exact computation is #P-hard. The paper mentions approximate randomized methods, but it does not analyze how using such approximations would affect the regret bound. This leaves the computational side of the result somewhat incomplete.
* **W3:** There is no experimental evaluation. Even for a theory paper, a small synthetic experiment comparing the simple $O(d^2)$ method with the center-of-gravity-based method would be helpful. It could also give some insight into how approximate center-of-gravity computation affects practical performance.
* **W4:** Some of the proofs, especially the lower-bound proof, are written quite compactly and are therefore somewhat hard to follow.

---

> ### Author Rebuttal · Authors · 2026-03-27
>
> We thank the reviewer for the thorough and careful review. We address each point below.
>
> ## **W1. The M-convex assumption is strong.**
>
> The reviewer's point is correct: we do not claim to establish finite regret for general online inverse optimization, only for the M-convex case. That said, as discussed in Section 1.1 “Motivation for M-Convex Action Sets” and the examples in Section 2.3, the M-convex case is rich in motivation and applications, and we believe our result is meaningful as a step toward the broader open question.
>
>
> ## **W2. Approximate center of gravity not analyzed.**
>
> Please see our response to Reviewer ef6R (W2) for a detailed analysis. In short, using the hit-and-run algorithm of Lovász and Vempala (2006), the approximate Grünbaum's theorem (Feldman et al. 2015, Lemma 5.6) guarantees that the volume shrinks by a constant factor per mistake, yielding $\mathbb{E}[R_T] = O(d\log d)$. This centroid approximation runs in $\tilde{O}(d^6 \log d \cdot \log T)$ time per round. We will add this analysis to the revision.
>
> ## **W3. No experimental evaluation.**
>
> We appreciate this suggestion. We have conducted a synthetic experiment directly addressing this weakness; please see our response to Reviewer ef6R (W1) for the details. In short, our algorithms achieve lower regret than ONS (Sakaue et al. 2025b) and OGD (Bärmann et al. 2020), confirming the benefit of exploiting the M-convex structure. Moreover, Alg 2's performance improves with more hit-and-run iterations, showing that better centroid approximation improves practical performance. We will include this experiment in the revision.
>
> ## **W4. Lower-bound proof is hard to follow.**
>
> Thank you for raising this point. The key idea is simple. Sakaue et al. (2025b, Theorem 5.1) establish $R_T = \Omega(d)$ using hard instances where each $X_t$ is an axis-aligned line segment. Intuitively, since each round reveals information about only one component of $w^*$, the learner must estimate all $d$ components independently, incurring $\Omega(d)$ total regret.
>
> The challenge is that these axis-aligned segments are not M-convex by themselves. However, one can introduce a dummy coordinate $x(0)$ and replace each $X_t$ with
>
> $$\left\lbrace(x(0), x(1), \ldots, x(d))\in\mathbb{Z}^{d+1} : (x(1), \dots, x(d)) \in X_t,\ x(0)=-\sum_{i=1}^d x(i)\right\rbrace,$$
>
> which is an M-convex set (Murota 2003, Section 6.1). This embeds the hard instances of Sakaue et al. (2025b) into the M-convex setting (with dimension $d+1$), yielding $R_T = \Omega(d)$ for the M-convex case as well.
>
> ## **Q1. What changes if $w^*$ has tied components?**
>
> Thank you for this important question. Our current presentation assumes that the components of $w^*$ are distinct, as stated in Assumption 2.2 (2). This assumption is mainly for expositional simplicity.
>
> In fact, it is not needed for the uncorrupted model. If $w^\ast$ has ties, then whenever the directed graph $([d], A\_t)$ develops a strongly connected component (SCC), Proposition 2.4 implies that all coordinates of $w^\ast$ in that SCC must have the same value. Thus, in the uncorrupted case, one can work with the condensation graph of SCCs rather than with individual coordinates, and the same proof idea can be carried out after contracting SCCs. We imposed distinctness only to simplify the presentation: explicitly maintaining SCCs would make the description more cumbersome and obscure the main idea.
>
> By contrast, in the corrupted model, distinctness in Assumption 2.2 (2) becomes important, because when $([d], A\_t)$ contains a cycle, our current argument interprets this as evidence of corruption rather than a tie. We agree that the present version does not make this subtle difference sufficiently explicit, and in the revised version, we will clarify this distinction more clearly. We thank the reviewer for drawing our attention to this point.
>
> ## **Q2. Why can Step 4 be solved in polynomial time with only a membership oracle for $X_t$?**
>
> Thank you for raising this question. The polynomial-time solvability follows from a known result for M-convex sets (more generally, jump systems); see Shioura and Tanaka (2007, "Polynomial-Time Algorithms for Linear and Convex Optimization on Jump Systems," Theorems 3.1 and 3.2).
>
> As an important and intuitive special case, consider when $X_t$ is a matroid. The standard greedy algorithm applies: sort the weights $\hat{w}_t$ in decreasing order in $O(d \log d)$ time, and add each element if doing so keeps the current set independent (one independence oracle call per element). This solves Step 4 in $O(d \log d)$ time plus $O(d)$ oracle calls. We will add this explanation to the revision.

---

> > ### Author Rebuttal · Reviewer_1zLJ · 2026-04-02
> >
> > Thank you for the thorough rebuttal. The authors have addressed most of my questions and clarified several concerns I raised. I appreciate the additional explanations and the planned revisions.
> >
> > My primary remaining concern is that the assumption of M-convexity is quite strong, which I believe limits the scope and significance of the results. For this reason, I will maintain my assessment of the significance as fair. Overall, the rebuttal was helpful, but I will keep my score unchanged.

---

### Official Review · Reviewer_qWN2 · 2026-03-12

**Soundness:** 4
**Presentation:** 4
**Significance:** 4
**Originality:** 4
**Overall Recommendation:** 5
**Confidence:** 5

**Summary:**

The paper studies online inverse linear optimization, where a learner sequentially estimates an agent's hidden objective vector by observing their optimal actions over changing feasible sets. Previous work has regret bounds that are exponentially large in the dimension $d$, or depend on horizon T, leaving open the existence of a regret bound only on dimension $d$. The authors partially resolve this by proving that when the agent's feasible sets are M-convex (a broad class that includes matroids) that a regret bound of $O(d \log d)$ is achievable.

This result is obtained by combining a structural characterization of optimal solutions on M-convex sets with a geometric volume argument based on order simplices.The authors also handle adversarial corruptions in up to $C$ rounds. By monitoring directed graphs induced by the feedback to adaptively detect cycles, the algorithm restarts when necessary and achieves a regret bound of $O((C+1)d \log d)$ without needing prior knowledge of $C$. Prior work has a $\Omega(d)$ lower bound for M-convex sets, so their upper bound is tight up to a log(d) factor.

**Compliance With Llm Reviewing Policy:**

Affirmed.

**Key Questions For Authors:**

- What is the relationship between the regret bound defined in Equation (2) and the bound on the estimation error for $w^*$, which is the standard goal in offline inverse optimization?
- Can this approach be adapted to a setting where, instead of observing the strictly optimal action $x_t$, the action $x_t$ is generated sequentially by an online learning algorithm? I see an interesting connection between your results and the problem of inverse online learning https://petriconi.org/assets/pdf/ARLET_2024.pdf, https://openreview.net/pdf?id=kQQ20pTbxI

**Limitations:**

Yes

**Strengths And Weaknesses:**

Strengths
- The authors provide a strong theoretical resolution to a known open problem by proving a regret bound that is polynomial only in $d$ for M-convex action sets.
- I enjoyed reading Section 4, where the local exchange property of M-convex sets is used to get pairwise inequality constraints on the hidden objective vector. By selecting the learner's estimate $\hat{w}_t$ as the center of gravity of the resulting polytope, any mispredicted action guarantees that the newly revealed constraint acts as a cutting plane through the centroid.
- They also handle corrupted feedback dynamically, removing the need to know the corruption level $C$ in advance by tracking the acyclicity of directed graphs.

Weakness
- Computing the exact center of gravity for the parameter estimates in Step 3 is #P-hard, how does using randomized approximation affect the O(d log d) regret?
- For Sec 2.2 and the result in Sec 5, how does regret bound change if we the corrupted vector is close to the true vector? You mention a O(d log T) factor from Sakaue et al, but is it better for your M-convex settting?

---

> ### Author Rebuttal · Authors · 2026-03-26
>
> We thank the reviewer for the detailed and insightful review.
>
> ## **W1. Approximate center of gravity and regret.**
>
> Please see our response to Reviewer ef6R (W2), which addresses this in detail. In short, using the hit-and-run algorithm of Lovász and Vempala (2006) and its guarantee formalized in Feldman et al. (2015, Theorem 5.7), the approximate Grünbaum's theorem (Feldman et al. 2015, Lemma 5.6) guarantees that the volume shrinks by a constant factor per mistake, yielding the same asymptotic regret bound of $\mathbb{E}[R\_T] = O(d\log d)$. The centroid approximation runs in $\tilde{O}\left(d^6\log d\cdot \log T\right)$ time per round.
>
> ## **W2. Regret bound when corrupted vectors are close to the true vector.**
>
> Our corruption model (following Krishnamurthy et al. 2021; Paes Leme et al. 2022, 2025; Gupta et al. 2025) counts the number of corrupted rounds $C$, regardless of how close each corrupted action is to optimal. Our bound $O((C+1)d\log d)$ thus grows with $C$ even if the observed corrupted actions are very close to the true optimal actions.
>
> As noted in Section 2.2, Sakaue et al. (2025) study a more flexible model where corruption is measured by the cumulative suboptimality $\Delta\_T = \sum\_t \langle w^{\ast}, x^{\ast}\_t - x\_t \rangle$, where $x^{\ast}\_t$ is the true optimal action and $x\_t$ is the observed (corrupted) action. They present a regret bound of
> $$
> R\_T = \sum\_t \langle w^{\ast}, x\_t - \hat{x}\_t \rangle = O(d\log T + \sqrt{d\Delta\_T \log T}),
> $$
> which degrades gracefully with $\Delta\_T$; however, their bound still has a $\log T$ factor. Thus, our bound is better when $T$ is large while $C$ is small, whereas the bound of Sakaue et al. (2025) can be better when $T$ and/or $\Delta\_T$ are small. Extending our approach to their corruption model while retaining a finite regret bound is an interesting open direction, as noted in Sections 2.2 and 7.
>
> ## **Q1. Relationship between regret (eq. 2) and estimation error for $w^{\ast}$.**
>
> Our regret $R\_T = \sum\_t \langle w^{\ast}, x\_t - \hat{x}\_t \rangle$ measures the cumulative gap between the agent's optimal action values and the values of the learner's recommended actions. This is the same metric used in prior work summarized in Table 1.
>
> On the other hand, the estimation error for $w^{\ast}$, e.g., $\\|w^{\ast} - \hat{w}\\|$, is a natural metric in the offline setting but faces a fundamental identifiability issue in our online inverse linear optimization setting. Since we only observe actions $x\_t$ from a linear forward optimization, even the scale of $w^{\ast}$ is unobservable. More importantly, components of $w^{\ast}$ that do not affect the choice of any observed optimal action $x\_t$ are entirely uninferable. Consequently, by the problem structure, $\\|w^{\ast} - \hat{w}\\|$ can be large in the worst case, no matter how well the learner performs. Our regret definition sidesteps this fundamental pessimism by evaluating $\hat{w}\_t$ solely on whether it induces near-optimal actions $\hat{x}\_t$, capturing the aspects of $w^{\ast}$ that are learnable from the observed feedback (see also Remark 2.1).
>
> That said, if one places additional assumptions on how the feasible sets $X\_t$ are generated—ensuring that every component of $w^{\ast}$ influences the choice of some optimal action $x\_t \in X\_t$—then achieving small regret may also imply small estimation error. Formalizing this connection is an interesting direction for future work.
>
> ## **Q2. Connection to reward estimation in inverse online learning.**
>
> We thank the reviewer for pointing us to these works and for drawing attention to this exciting connection.
>
> As discussed in our response to Q1, our regret evaluates $\hat{w}\_t$ by action quality rather than by estimation error $\\|w^{\ast} - \hat{w}\_t\\|$. Since our method is not designed to minimize estimation error, it does not have an immediate implication for reward estimation in the inverse RL/bandit setting where $x\_t$ is generated by an online learning algorithm. (Interestingly, in the robust regret bound of Sakaue et al. (2025), $\Delta\_T$ can be seen as the regret of the online learning algorithm generating $x\_t$; this may bring another angle to connect their work with inverse online learning.)
>
> That said, we find the potential connection intriguing. Guha et al. (2024) obtain estimation error bounds by placing a smoothness assumption on the action sets. By analogy with the Q1 discussion—where structural assumptions on how $X\_t$ is generated could potentially bridge regret and estimation error—the degree to which $w^{\ast}$ influences actions taken in forward optimization may also be tied to structural properties of the action sets, like the smoothness assumption. If so, developments in online inverse linear optimization and reward estimation in inverse RL/bandit could benefit each other. We plan to discuss this connection in the revised version, citing the suggested works.

---

> > ### Author Rebuttal · Reviewer_qWN2 · 2026-04-08
> >
> > Thank you for the detailed rebuttal, please include this approximate center of gravity and regret discussion in the final version. As for the score I will keep my original rating of Accept.

---

### Official Review · Reviewer_ef6R · 2026-03-23

**Soundness:** 4
**Presentation:** 4
**Significance:** 4
**Originality:** 4
**Overall Recommendation:** 5
**Confidence:** 1

**Summary:**

This paper studies online inverse linear optimization with M-convex feasible sets. The authors solved an open problem whether a finite regret bound polynomial in d is achievable.

**Compliance With Llm Reviewing Policy:**

Affirmed.

**Final Justification:**

I would like to maintain my score of 5 (accept).

**Key Questions For Authors:**

See the weaknesses.

**Limitations:**

yes.

**Strengths And Weaknesses:**

The theoretical analysis looks solid and rigorous. The paper is well presented.

Weaknesses:

This paper is purely theoretical and no experimental evaluation is provided. would be great to add at least numerical experiments using synthetic data.

the authors mentioned "the exact computation of the center of gravity is #P-hard" and suggested approximate algorithms. However, how does the approximate error affect the regret bound? There is no analysis on this.

---

> ### Author Rebuttal · Authors · 2026-03-26
>
> We appreciate the reviewer's positive assessment and constructive feedback.
>
> ## **W1. No experimental evaluation.**
>
> We conducted a synthetic experiment on time-varying uniform matroids: $d=20$ items, $k=10$ randomly unavailable items per round, $X_t$ = {all $m$-subsets ($m=5$) of the $d-k=10$ available items}. We set $T=5000$ and run five independent trials. We compare our $O(d^2)$-regret algorithm (Alg 1) and $O(d\log d)$-regret algorithm (Alg 2) using $N$ hit-and-run iterations (Lovász and Vempala, 2006. "Hit-and-run from a corner") for centroid approximation (see our response to W2 for the analysis), against ONS (Sakaue et al. 2025b) and OGD (Bärmann et al. 2020).
>
> | Algorithm | Regret (mean ± CI95) | Time/round |
> |---|---|---|
> |Alg 1|1.640 ± 0.339|11 μs|
> |Alg 2 ($N=10^2$)|1.677 ± 0.260|23 μs|
> |Alg 2 ($N=10^3$)|1.491 ± 0.156|120 μs|
> |Alg 2 ($N=10^4$)|**1.201 ± 0.382**|1.1 ms|
> |ONS|3.510 ± 0.776|70 μs|
> |OGD|4.848 ± 1.105|4 μs|
>
> Our algorithms achieve lower regret than ONS and OGD, confirming the benefit of exploiting the M-convex structure. The performance of Algorithm 2 improves with more hit-and-run iterations, showing that better centroid approximation improves practical performance. We will include this experiment in the revision.
>
> ## **W2. How does approximate center-of-gravity computation affect the regret bound?**
>
> We thank the reviewer for raising this important point. In the paper, we prioritized simplicity of the regret analysis and did not elaborate on the approximation of the center of gravity, following the same standard simplification as in Gollapudi et al. (2021). Based on the analysis below, we will add a detailed treatment of the computational aspect to the revised version to make this point clearer.
>
> **Prerequisites for Feldman et al. (2015, Theorem 5.7).** Recall that the $O(d\log d)$-regret algorithm (Section 4) selects $\hat{w}_t$ as the centroid of
>
> $$P_t=\\{w\in[0,1]^d : w(i)\ge w(j)\ \forall (i,j) \in A_t\\}.$$
>
> To approximate the centroid in polynomial time, we use the hit-and-run algorithm (Lovász and Vempala, 2006) and analyze it based on Theorem 5.7 in Feldman et al. (2015). The theorem requires (i) a membership oracle for $P_t$ and (ii) a starting interior point $q$ with known ball containment
> $R_0B_2^d\subseteq P_t-q\subseteq R_1B_2^d$.
> Both are readily available: the oracle checks box and order constraints in $O(d^2)$ time, and $q$ is obtained by topological-sorting $([d], A_t)$ to get $\pi$ and setting $q(\pi(k))=1-k/(d+1)$, giving $R_0=1/((d+1)\sqrt{2})$ and $R_1=\sqrt{d}$ (by direct calculation), hence $R_1/R_0=O(d^{3/2})$.
>
> **Approximate centroid guarantee.** With the above prerequisites, Feldman et al. (2015, Theorem 5.7) guarantee that, for any target accuracy $\tau > 0$ and failure probability $\delta > 0$, with probability $\ge 1-\delta$ one can compute $\hat{w}\_t$ satisfying
>
> $$\\|\hat{w}\_t -z(P\_t)\\|_{E\_{P\_t}}\le\tau$$
>
> in $\tilde{O}(d^4\log d\cdot\log(1/\delta)/\tau^2)$ membership-oracle calls. Here, $z(P\_t)$ denotes the exact centroid, $\Sigma(P\_t) = \mathbb{E}\_{w\sim{\rm Unif}(P\_t)}[(w-z(P\_t))(w-z(P\_t))^\top]$ is the covariance matrix, $E\_{P\_t} = \\{u : u^\top\Sigma(P\_t)^{-1}u \le 1\\}$ is the inertial ellipsoid, and $\\|\cdot\\|\_{E\_{P\_t}} \coloneqq \sqrt{(\cdot)^\top \Sigma(P\_t)^{-1}(\cdot)}$ is the induced norm.
>
> **Volume shrinkage with approximate centroid.** We then check that replacing the exact centroid by $\hat{w}\_t$ preserves the constant-factor volume shrinkage per mistake, i.e., an approximate version of our Lemma 4.1. In a mistake round ($x\_t \ne \hat{x}\_t$), there exists $(i,j)\in A\_{t+1}\setminus A\_t$ with $\hat{w}\_t(i) < \hat{w}\_t(j)$, so $\hat{w}\_t$ lies in $H\_{ij}^-=\\{w : w(i) \le w(j)\\}$, opposite to the new feasible halfspace $H\_{ij} = \\{w : w(i) \ge w(j)\\}$. Applying Feldman et al. (2015, Lemma 5.6; the approximate Grünbaum's theorem, mentioned in footnote 4) to $H\_{ij}^-$ yields
>
> $$\mathrm{Vol}(P\_t \cap H\_{ij}^-)\ge\left(1/e -\tau\right)\mathrm{Vol}(P\_t),$$
>
> and since $P\_{t+1}\subseteq P\_t\cap H\_{ij}$, we have
>
> $$\mathrm{Vol}(P\_{t+1})\le\left(1-1/e+\tau\right)\mathrm{Vol}(P\_t).$$
>
> **Regret bound and runtime.** For any constant $\tau < 1/e$, e.g., $\tau=1/(2e)$, the volume shrinks by a constant factor per mistake. Combined with $\mathrm{Vol}(P\_T) \ge 1/d!$ (since $P\_T$ contains the order simplex of $w^*$), the number of mistakes is $O(d\log d)$, giving total regret $O(d\log d)$ (each mistake contributes $O(1)$ by Assumption 2.1). For the high-probability statement, set $\delta=1/T^2$ per round and take the union-bound over all $T$ rounds. Then, we have $R\_T=O(d \log d)$ w.p. $\ge 1 - 1/T$ and $R\_T=O(T)$ w.p. $\le 1/T$, so $\mathbb{E}[R\_T]=O(d\log d)$. The per-round oracle-call count is $\tilde{O}\left(d^4\log d\cdot \log T\right)$, each of which takes $O(d^2)$ time. Therefore, the approximate centroid computation takes $\tilde{O}\left(d^6\log d\cdot \log T\right)$ time per round.

---

> > ### Author Rebuttal · Reviewer_ef6R · 2026-04-02
> >
> > The authors addressed both my questions (about the synthetic experiments and additional analysis).

---

### Decision · Program_Chairs · 2026-04-30

**Decision:**

Accept (regular)

**Comment:**

This paper investigates the online inverse linear optimization problems over M-convex feasible sets, and establishes an $O(dlogd)$ regret bound that is optimal up to a logarithmic order. Overall, the theoretical results are solid, and the developed technical tools would be potentially helpful for investigating the general online inverse linear optimization problems. The reviewers' comments have been adequately addressed during the rebuttal period. I recommend acceptance.